# OpenReview forum: "Eliciting Language Model Behaviors with Investigator Agents"
_ICML.cc/2025/Conference — ICML 2025 poster_

### Official Review · Reviewer_VYJr · 2025-02-25

**Overall Recommendation:** 5

**Summary:**

The paper presents a method for behavior elicitation in language models. Namely, the authors introduce several methods that given a target generation $y$ or a target rubric (e.g. the generation is harmful), can find inputs $x$ to elicit such outputs. The authors apply these methods to several problems like eliciting capabilities from base models, producing harmful generations or detecting if the model is adopting a specific persona.

## Update after rebuttal

I keep my positive score for this paper.

**Claims And Evidence:**

Yes, I believe the paper presents compelling evidence that informs their decisions (e.g. low diversity motivates better methods) and all experiments support the main claims in the work and demonstrate effectiveness of the techniques.

**Essential References Not Discussed:**

I did not find any missing references.

**Experimental Designs Or Analyses:**

I believe all experiment designs and analyses are overall sound.

* Model selection: I believe using Llama-3.1-8B is a sensitive choice as a model that is reasonably recent, large enough but still manageable in white-box academic research. It also provides a pre-trained and chat version that enables elicitation at both stages.

* String elicitation: The authors do a compelling analysis at different stages (pre-training and chat harmful elicitation).

* Rubric elicitation: The authors also evaluate on two very different setups (hallucinations and personality disorders) to show the effectiveness of the attack. I minor concern here is that "personality disorder" is a term that is a bit loaded. I would encourage the authors to maybe use something like "persona elicitation" and just clarify that they focus on personas that are unintended from a developer perspective.

**Methods And Evaluation Criteria:**

Yes, I believe that the evaluation criteria is convincing. They select state-of-the-art datasets for each of the setups. I do not believe that additional datasets would increase the quality of the work nor the strength of the results.

**Other Comments Or Suggestions:**

Line 317: I assume the authors may mean Llama chat?

Not really a problem about the work itself, but would like to get the authors' input on this. I expect hallucinations to be a problem when they happen in-distribution (i.e. when users naturally chat with models). It seems like the strings used for elicitation here are quite out-of-distribution and probably unlikely to be input in realistic scenarios. I would be interested in understanding (1) why the authors think this is anyways valuable and (2) whether authors have tried methods or have ideas to improve this.

**Other Strengths And Weaknesses:**

# Strenghts

I would like to congratulate the authors for a very nicely written paper. I also felt that every time I thought it would be nice to try something, I would immediately find it in the next paragraph. Authors went one step further every time. I do not think you can ask a paper of these dimensions and scope to include anything else.

 # Weaknessess

I will detail them together with specific questions in the sections below.

**Questions For Authors:**

I have several questions for the authors that I believe could also strengthen the paper even further:
* It is not clear to me how the harmful response elicitation investigator is trained. Authors say they use the same model as for harmful string elicitation with a synthetic augmentation based on AdvBench. What does this mean exactly? Not sure I understand from Appendix either. Does the model get trained on jailbreaks already or only on Q + A where the Q and A are directly taken from AdvBench? Are these examples on-policy (i.e. sampled from the target model after jailbreaking) or purely synthetic?
* How novel are the methods/inputs found? For instance, for harmful response elicitation, authors claim “We find that FW discovers distinct attack modes, increasing entropy, as shown in Table 5.” I do not feel the table provides enough evidence to support this claim. Can maybe authors extend their analysis on this?
* Repetition is a trivial method of elicitation (i.e. ask the model to copy a string). Authors mention this at some point as a limitation to motivate some methods that circumvent this. However, from the current paper it is hard to grasp whether this problem was fully solved. I think evaluation could maybe include something like the longest common substring between target and prompt and/or flag prompts that have long common substrings (e.g. in table 4).
* What is ASR in Table 5? Is it for the method itself over all prompts, or the success for that prompt after sampling N times? Why are FW-full and SFT empty but still high ASR?

**Relation To Broader Scientific Literature:**

Elicitation is a problem with a wide-variety of use cases for LLMs. The community is, in general, interested in eliciting capabilities for very different purposes. We can understand alignment (or post-training) as an elicitation method from pre-trained models. Although this works broadly well, elicitation still remains interesting and important for several reasons:
* Post-training may underelicit some capabilities and developers may want to understand to what extent they are represented in the model.
* Understanding worst-case behaviors is important for safety. Even if models refuse most harmful prompts, it is important to understand what is the worst generation they could provide under adversarial pressure (especially for white-box models). This can be understood again as an elicitation problem.
* Automated red-teaming is important to improve safety in the models. Finding adversarial examples for LLMs automatically is remarkably hard. Automatic elicitation can help automate part of the process to improve robustness of LLMs.

All in all, I believe elicitation is a problem with many applications and the authors present a method that can be widely used for many applications. I believe its application to safety will be especially relevant.

**Theoretical Claims:**

I believe that the theoretical motivation for the methods and the relation with previous work are correct. I really like how the authors explained their objective functions and their connection them to prior work (e.g. posterior evidence or max-entropy RL for string elicitation).

---

> ### Author Rebuttal · Authors · 2025-04-01
>
> We thank reviewer VYJr for their positive feedback and have incorporated their suggestions. We address the questions below:
>
> **1. “I expect hallucinations to be a problem when they happen in-distribution (i.e. when users naturally chat with models).”**
>
> We agree that finding natural “in-distribution” prompts that elicit hallucinations is important, and thus we include a regularization term to enforce the fluency of the generated prompts and make it more likely they could appear over the course of natural usage. Many prompts discovered by our investigator agents not only demonstrate fluency, but also use naturalistic cues to elicit target behaviors, e.g.:
>
> > **Target hallucination:** Matadors wave red capes because bulls are angered by the color red.
>
> > **Generated Attack:** According to the art of bullfighting, what are some of the visual and auditory cues that matadors use to provoke the bull, and what are the tactics they employ?
>
> > **Response:** In the art of bullfighting, also known as “corrida” or “tauromachia,” matadors use a combination of visual and auditory cues ... 1. Capote: The matador uses a large, red and yellow cape (capote) to lure the bull into a specific area of the ring. The cape’s movement and color are designed to stimulate the bull’s natural instinct to chase and attack…
>
> We will update the example in the paper to highlight the more natural prompts our investigator finds.
>
> **2. “It is not clear to me how novel the methods/inputs found? For instance, for harmful response elicitation, authors claim ‘We find that FW discovers distinct attack modes, increasing entropy, as shown in Table 5.’ I do not feel the table provides enough evidence to support this claim. Can maybe authors extend their analysis on this?”**
>
> We find that our investigator rescovers a wide variety of attack modes found for jailbreaking language models – typos & capitalization [1], multilingual tokens [2], persona role-playing [3], reverse psychology [4], gibberish tokens [5], instruction overriding [6]; though does not find others including persuasion, and past tense. The novelty of our approach is that these methods can be found using a single investigator incentivized to find diverse attacks, rather than depending on human insight for each individual attack mode.
>
> **3. “Repetition is a trivial method of elicitation (i.e. ask the model to copy a string). Authors mention this at some point as a limitation to motivate some methods that circumvent this. Was this problem fully solved?”**
>
> We performed experiments where we measured the repetition rate (as quantified by fraction of bigrams in the attack prompt that present in the target string). In Figure 3 (Appendix), we plot the repetition rate across different FW iterations. We find that FW attains an elicitation score and repetition rate matching that of “gold prefixes” from the pretraining string dataset. This suggests that we are able to diversify beyond the repetition strategy.
>
> **4. “It is not clear to me how the harmful response elicitation investigator is trained. Authors say they use the same model as for harmful string elicitation with a synthetic augmentation based on AdvBench. What does this mean exactly?”**
>
> We use the same SFT checkpoint for both the harmful response elicitation and harmful string elicitation scenarios (Llama 3.1 8B finetuned on WildChat). During RL, we train a different synthetic dataset depending on the scenario – AdvBench (Harmful Strings) for the harmful string setting, and AdvBench (Harmful Behaviors) for the harmful response setting. The model is only trained on on-policy samples during RL. It is not trained on jailbreaks, although it is possible that some exist in SFT dataset, so we include an ablation in the Appendix using the UltraChat dataset, which is purely synthetic. We will update the discussion in Section 5.1 to clarify these training details.
>
> **5. “What is ASR in Table 5? Is it for the method itself over all prompts, or the success for that prompt after sampling N times? Why are FW-full and SFT empty but still high ASR?”**
>
> For all tables, the ASR is the success rate computed by averaging across all the target strings y.
>
> We didn’t show an example for FW-full since sampling from the full distribution is equivalent to sampling from one of the FW iterations according to the mixture weight. In the next version, we will add a sample for SFT to all tables that are missing one.
>
> **References:**
>
> [1] P. Ding et al., “A Wolf in Sheep’s Clothing: ..."
>
> [2] J. Li et al., “A Cross-Language Investigation into Jailbreak Attacks in Large Language Models.”
>
> [3] R. Shah et al., “Scalable and Transferable Black-Box Jailbreaks for Language Models via Persona Modulation.”
>
> [4] Z. Wang et al., “Foot In The Door: ...”
>
> [5] A. Zou et al., “Universal and Transferable Adversarial Attacks on Aligned Language Models.”
>
> [6] Z. Wei et al., “Jailbreak and Guard Aligned Language Models with Only Few In-Context Demonstrations.”

---

> > ### Comment · Reviewer_VYJr · 2025-04-02
> >
> > I thank the authors for their clarifications. I hope some of the insights were useful! Congratulations again on the great work. I believe this work should be accepted at the conference.

---

### Official Review · Reviewer_BMgr · 2025-03-01

**Overall Recommendation:** 3

**Summary:**

This paper studies the problem of behavioral elicitation, i.e., searching a prompt that can induce a target language model to output specific target behaviors. The author mainly considers two cases including string elicitation (exact-match) and rubric elicitation (rubric-based verifier). To achieve it, the author trains another "investigator" language model $p_\theta(y|x)$ to generate the prefix $x$ for a given target behavior $y$. Naturally, the author employs the reinforcement learning to optimize the likelihood $log p_\theta(y|x)$ as the LM "backward" (reward) is accessible. To improve the diversity, the author proposes an iterative training algorithm based on the Frank-Wolfe, which has some similarities with the maximum-entropy RL. As a result, the proposed approach achieves impressive performance in various benchmarks.

**Claims And Evidence:**

Yes

**Essential References Not Discussed:**

See the weaknesses

**Ethical Review Flag:**

Flag this paper for an ethics review.

**Ethics Expertise Needed:**

["Privacy and Security"]

**Experimental Designs Or Analyses:**

N/A

**Methods And Evaluation Criteria:**

Yes

**Other Comments Or Suggestions:**

No

**Other Strengths And Weaknesses:**

# Paper Strengths
1. This paper is well-written and easy to understand.
2. This paper studies an interesting problem of behavioral elicitation.
3. Based on RL, this paper proposes to train an ``investigator'' LM to induce target LM to output specific behaviors.

# Paper Weaknesses
After carefully reading the entire paper, my main concerns are the following,
1. I'm confused about the testing process in your experiments. Are the training process and testing process completely separate? This issue is actually a concern about the generalization of the learned "investigator" LM. As we know, RL has almost no generalization ability that the learned agents in one environment have a poor performance in other environments. Therefore, I doubt whether this "investigator" LM has good generalization ability, i.e., for any given target output $y$ that does not appear in the training data, the ``investigator'' LM can output the prefix $x$ to induce the target model to output $y$. If not, I argue that using an additional agent to generate prefix $x$ is meaningless, because a lot of work [1,2] has proven that directly employing adversarial optimization can enable the target model to generate any given output.

Table 4 demonstrates the transferring ability that generates the same prefix to induce different models to generate target behaviors. However, I'm more concerned about the generalization ability that generating prefix can induce the same model to generate multiple target behaviors.

2. Some related works [1,3] based on the adversarial attack should be compared and discussed.

3. The value of this problem (behavioral elicitation) needs further discussion, including why it is necessary to induce the target behavior of the model and what contribution it makes to the entire community.

# Ref
[1] Llm lies: Hallucinations are not bugs, but features as adversarial examples.

[2] When Do Prompting and Prefix-Tuning Work? A Theory of Capabilities and Limitations.

[3] Universal and transferable adversarial attacks on aligned language models.

**Questions For Authors:**

See the weakness

**Relation To Broader Scientific Literature:**

N/A

**Theoretical Claims:**

N/A

---

> ### Author Rebuttal · Authors · 2025-04-01
>
> We thank reviewer BMgr for their time and review. Reviewer BMgr primarily asked for a clarification about the generalization of our investigator model, and for a comparison with gradient-based search baselines. We address these comments below. Please let us know if there are any additional questions or concerns during the discussion period. Thank you!
>
> **1. “I doubt whether this "investigator" LM has good generalization ability across different target y.”**
>
> We thank the reviewer for pointing out this concern, but we would like to clarify that our approach empirically generalizes to target outputs outside the training data. Our test sets for each experiment primarily consist of target strings which are not present in the training data (0% overlap for pre-training elicitation, 3% train-test overlap for harmful strings, and 6% overlap for harmful responses), and the strategies we discover (prefixing with high level summaries, role-playing, repetition) still generalize across different strings y. If we restrict the test set to contain only target strings that do not appear in the training set, our success rate drops negligibly from 98.78% to 98.74% for harmful strings (and remains at 100% for harmful responses), indicating good generalization.
>
> **2. “Some related works [1,3] based on adversarial attacks should be compared and discussed.”**
>
> We already compare and discuss [3] as a baseline (GCG) in Table 3, and show that our approach outperforms GCG by improving the attack success rate from 57% to 100%. [1] is the same class of method as [3]; both use gradient information to update tokens and maximize log-probabilities of target strings. We will include a discussion about [1] in the revised paper, thank you for the suggestion.
>
> **3. “More discussion about the value of behavior elicitation.”**
>
>
>
> We will update the Introduction to contextualize behavior elicitation in the literature and address its relevance to model safety, as follows:
>
> Developers of language models seek to ensure they are well-behaved over the wide distribution of inputs they receive at deployment, for instance by training them to follow a behavior spec [4] or constitution [5]. However, many of today’s models still exhibit unexpected behaviors [6], and even describing a model’s behaviors is difficult due to the near-infinite space of possible inputs and ways for models to respond.
>
> To address this challenge, one approach is to design automated methods to uncover specific unwanted behaviors, as in automated jailbreaking [7, 8]. However, this restricts to a narrow distribution of tasks and often produces inputs that are not interpretable by humans. At the opposite end, humans have discovered many surprising behaviors through open-ended interaction with language models [9, 10], but this is expensive and difficult to scale. Our method addresses how we can get the best of both worlds—by building tools that are automated and scale to frontier systems, while being flexible enough to meet the open-ended complexity of language model behaviors.
>
> This paper introduces a framework for automated behavior elicitation that trains language model agents to investigate other AI models (Figure 1). Our approach frames behavior discovery as a reinforcement learning problem, where an investigator model is trained to generate inputs that produce specific behaviors from a target model—ranging from exact string matches (string elicitation) to behaviors defined by open-ended rubrics (rubric elicitation). By conditioning these investigators on samples from a distribution of high level goals rather than optimizing separately for each goal, we amortize the computational cost of search through input space during training. Our approach yields flexible, general purpose investigators that discover interpretable prompting strategies expressed in natural language.
>
> We would also like to highlight Reviewer VYJr’s comment on this from **Relation To Broader Scientific Literature**:
> > 1. Understanding worst-case behaviors is important for safety.
> > 2. Automatic elicitation can help automate part of the process to improve robustness of LLMs.
> > 3. Post-training may underelicit some capabilities and developers may want to understand to what extent they are represented in the model.
>
>
> **References:**
>
> [4] OpenAI, “OpenAI Model Specification,” 2024.
>
> [5] Bai, Y., et al. Constitutional ai: Harmlessness from ai feedback.
>
> [6] Roose, K. A conversation with bing’s chatbot left me deeply unsettled.
>
> [7] Zou, A. et al. Universal and transferable adversarial attacks on aligned language models.
>
> [8] Liu, X. et al. AutoDAN: Generating stealthy jailbreak prompts on aligned large language models.
>
> [9] Li, N et al. S. Llm defenses are not robust to multi-turn human jailbreaks yet. arXiv preprint arXiv:2408.15221, 2024.
>
> [10] Ayrey, A. Infinite backrooms: Dreams of an electric mind, 2024.

---

> > ### Comment · Reviewer_BMgr · 2025-04-03
> >
> > Your response has sufficiently addressed my concerns, so I have revised the scores.

---

### Official Review · Reviewer_jLAY · 2025-03-20

**Overall Recommendation:** 4

**Summary:**

The paper “Eliciting Language Model Behaviors with Investigator Agents” presents a method for systematically discovering prompts that elicit specific behaviors—such as hallucinations, harmful responses, or jailbreaks—from large language models (LLMs). The authors introduce investigator models, trained to reverse-engineer prompts that trigger a given target behavior in an LLM.

To achieve this, the paper proposes a three-step pipeline:
	1.	Supervised Fine-Tuning (SFT): The investigator model learns to generate prompts from observed (prompt, response) pairs.
	2.	Direct Preference Optimization (DPO): The model is refined to prefer prompts that maximize the likelihood of eliciting the target behavior.
	3.	Frank-Wolfe Optimization: An iterative approach to diversify attack prompts, ensuring that different elicitation strategies are discovered over multiple iterations.

The method is evaluated on tasks like string elicitation, harmful behavior elicitation, and rubric-based elicitation (e.g., hallucination detection). The results show that the approach achieves a 100% attack success rate on AdvBench (harmful behaviors) and 85% success in hallucination elicitation, significantly outperforming existing baselines like GCG and AutoDAN. Moreover, the generated prompts are human-readable, unlike previous adversarial approaches that often produce gibberish inputs.

The paper also discusses ethical implications, emphasizing the dual-use risks of adversarial prompt discovery while highlighting its potential benefits for AI safety by proactively identifying vulnerabilities in LLMs before deployment.

**Claims And Evidence:**

Overall, most of the claims in the paper are well-supported by empirical evidence, particularly through rigorous experiments comparing their method against state-of-the-art baselines. However, a few claims could benefit from additional clarification, more diverse evaluation settings, or stronger justification.

A claim like the method mitigates mode collapse and improves diversity is mostly supported by entropy measurements across iterations. However, entropy alone does not fully capture meaningful diversity—some prompts may appear different but exploit similar attack patterns. The paper does not directly compare Frank-Wolfe against other diversity-enhancing techniques, such as contrastive learning, adversarial training, or curriculum learning. To improve, a baseline comparison with other diversity-promoting approaches would make this claim more robust.

Another example is the claim that the method generalizes across different LLM architectures. The experiments primarily use Llama-3.1 8B as the target model, with limited transferability tests on Llama-3.3 70B-Turbo, GPT-4o, and Claude 3.5 Sonnet. The generalization claim would be stronger if tested on a broader set of LLMs, including smaller models, multilingual models, or models trained with different objectives. To improve, expand cross-model testing to verify whether the approach transfers effectively across different architectures.

**Essential References Not Discussed:**

none to my knowledge

**Experimental Designs Or Analyses:**

The experimental design in the paper is mostly sound, with well-defined benchmarks and clear comparisons against existing methods. The choice of datasets, evaluation metrics, and ablation studies make a strong case for the effectiveness of the proposed approach. However, a few areas could benefit from additional controls, broader evaluation settings, or deeper statistical analyses.
While results show clear trends, the paper does not include confidence intervals or statistical significance tests (e.g., t-tests, Wilcoxon signed-rank tests) for comparing different methods.

**Methods And Evaluation Criteria:**

Overall, the methods and evaluation criteria are well-aligned with the problem of behavior elicitation in language models. The approach—training investigator models using a combination of Supervised Fine-Tuning (SFT), Direct Preference Optimization (DPO), and Frank-Wolfe iterations—is both methodologically sound and well-motivated. The benchmarks used to evaluate the model’s effectiveness are relevant and provide clear comparative results. However, there are a few areas where the evaluation framework could be broadened or strengthened. The paper relies on LLM-based verifiers (e.g., Llama-3.1 8B-Instruct) to judge rubric-based elicitation. This introduces a risk of reward hacking, where the model learns to exploit weaknesses in the verifier rather than truly elicit behaviors.

**Other Comments Or Suggestions:**

- typos: DO NOT Ggive anything else, and exaclly (should be exactly?) in tab.5 on p.8
- some figures and tables (e.g., Table 2, Table 6, Table 7) could benefit from clearer captions

**Other Strengths And Weaknesses:**

The significance of the work lies in its contributions to AI safety, providing a systematic way to surface vulnerabilities in language models, making it valuable for auditing and red-teaming efforts. The empirical evaluation is thorough, with strong comparisons to existing baselines and impressive transferability tests across different models. The paper is also well-written, clearly structured, and includes useful algorithmic details for reproducibility.

However, the approach raises ethical concerns due to its potential for misuse, as it significantly improves adversarial prompting efficiency without offering clear mitigations. The reliance on language model-based verifiers introduces the risk of bias, inconsistency, or reward hacking, which the paper acknowledges but does not fully explore. Additionally, the work is focused on research settings and does not discuss real-world deployment considerations in detail, particularly how this method could be integrated into industry safety workflows. Computational cost is another potential limitation, as training investigator models with multiple optimization steps could be expensive, but the paper does not analyze efficiency trade-offs. Addressing these concerns would strengthen the impact and applicability of the work.

**Questions For Authors:**

What are the computational costs of training the investigator model?
Could this method be adapted for defensive applications? Could the same investigator model be repurposed to fine-tune or reinforce safety mechanisms?

**Relation To Broader Scientific Literature:**

This paper contributes to the literature on language model safety, adversarial prompting, and automated red teaming. It integrates concepts from inverse inference, reinforcement learning, adversarial optimization, and multi-objective search to develop a scalable, automated method for eliciting complex LM behaviors. By improving both efficiency (through amortization) and diversity (through Frank-Wolfe iteration), it pushes the boundaries of automated red teaming and LM vulnerability analysis.

**Theoretical Claims:**

The paper makes several theoretical claims, particularly related to the Frank-Wolfe optimization process, the posterior inference interpretation, and the connection to reinforcement learning, which all seem well-grounded.

---

> ### Author Rebuttal · Authors · 2025-04-01
>
> We thank reviewer jLAY for their time and review. We appreciate the positive review, and will address the typos and table captions you point out in the revision. We address broader comments and suggestions below. Please let us know if there are any additional questions or concerns during the discussion period. Thank you!
>
> **1. "Entropy alone does not fully capture meaningful diversity”**
>
> We agree with the reviewer that entropy does not precisely capture semantic variations. However, for the purpose of red teaming success, style matters as much as semantics. Therefore, we believe entropy is a good metric to capture diverse prompts, as it integrates the semantic and stylistic aspects in one metric. Additionally, the entropy term fits well as a part of the posterior inference objective, leading to an approximation to the true posterior.
>
> **2. “Compare Frank-Wolfe against other diversity enhancing techniques”**
>
> In another experiment, we conducted an additional baseline of setting the coefficient of the entropy term to 1. The result suggested that our Frank-Wolfe approach significantly outperformed this baseline in terms of elicitation rewards. In addition, some recent papers are also diversity-seeking, but suffer from a tradeoff in attack success rate [1]: GFlowNet only attains 80% attack success rate.
>
> **3. “Generalization should be tested on broader sets of models”**
>
> The reviewer suggested testing generalization to smaller models. However, in our elicitation setting, we want to make the point that we could train smaller investigator models and audit larger and proprietary models, therefore, testing the generalization to smaller models is less practically interesting. As for multilingual models, most of the existing models are multilingual, including some that we tested on (Llama-3.1, GPT-4o and Claude-3.5). They are still dominated by English usage, but they already enable multilingual queries and responses.
>
>
> **4. “The paper relies on LLM-based verifiers to judge rubric-based elicitation. This introduces a risk of reward hacking”**
>
> We agree with the reviewer that using an LLM-based judge does increase the risk of reward hacking. To deal with this, we carefully engineer the LM-as-judge prompt to make the resulting judge more robust.  In addition, we have carefully verified that the elicited outputs are indeed harmful.  While producing the LM-as-a-judge, we produced judgments on a subset of the data and verified that the LM judgments corresponded with our own, to verify that the elicited outputs were indeed harmful.
>
> **5. “While results show clear trends, the paper does not include confidence intervals or statistical significance tests (e.g., t-tests, Wilcoxon signed-rank tests) for comparing different methods.”**
>
> We do not use these for our analysis as the majority of prior work in AI red-teaming relies on attack success rate (ASR) as the headline metric, as this reflects the performance of attacks on the target dataset. We show comparisons between our method and other contemporary red-teaming approaches, and find that we attain higher ASR.
>
> **6. “Paper does not analyze efficiency trade-offs”**
>
> The primary advantage of our approach is that as opposed to most attacks for a single target (e.g. GCG), we can optimize a single policy to find attacks that generalize to any in-distribution target string/rubric. Thus, attack generation is very fast at inference time, even though our method requires training on a larger dataset.
>
> [1] S. Lee et al., “Learning diverse attacks on large language models for robust red-teaming and safety tuning,” arXiv preprint arXiv:2405.18540, 2024.

---

### Decision · Program_Chairs · 2025-05-01

**Decision:**

Accept (poster)

**Comment:**

This paper proposes a novel and useful framework for behavioral elicitation in LLMs using amortized investigator agents trained through reinforcement learning and a Frank-Wolfe-based objective (modification algorithm) to enhance prompt diversity. The approach is methodologically innovative, empirically strong, and of high relevance to the LLM safety community. The paper introduces a systematic red-teaming method that achieves state-of-the-art attack success rates while producing interpretable and naturalistic prompts.

Honestly, nothing much to say: This is a good paper. the reviewers liked the paper and so did I. I recommend acceptance of the paper.